# Paediatric oncology nursing education and training programmes: a scoping review protocol

Maureen Daisy Majamanda ![ORCID],[1,2,3] Felix Chisoni ![ORCID],[4] Apatsa Selemani ![ORCID],[2,4,5] Irene Kearns ![ORCID],[3] Johanna Maree ![ORCID] [3]

¹Child Health Nursing, Kamuzu University of Health Sciences, Blantyre, Malawi
²Consortium for Advanced Research Training in Africa, Nairobi, Kenya
³Nursing Education, University of the Witwatersrand, Johannesburg, South Africa
⁴Library, Kamuzu University of Health Sciences, Blantyre, Malawi
⁵School of Public Health, University of the Witwatersrand, Johannesburg, South Africa

**Correspondence to**
Maureen Daisy Majamanda;
mdmajamanda@kuhes.ac.mw

## ABSTRACT

**Introduction** The care of children with cancer is a highly specialised field which requires well-educated, trained and dedicated nurses to provide high-quality care. In low/middle-income countries, the survival rate of children with cancer is low as compared with that of high-income countries due to the limited number of specialised oncology healthcare professionals, especially nurses. To address this problem, a number of paediatric oncology education and training programmes have been developed for nurses. The objective of this scoping review is to describe the existing literature focusing on paediatric oncology nursing education and training programmes; to map the content, delivery methods, duration and mode of assessment.

**Methods** The review will include articles published in English, from 2012 to 2022, that describe a paediatric oncology nursing education programme, from any setting. The review will follow Joanna Briggs Institute methodology for scoping reviews guidelines. A systematic search of literature will be performed in CINAHL, Dimensions, Embase, PubMed and Scopus. A two-stage standardised screening process will be employed to evaluate eligibility of the articles. All abstracts that will be considered relevant will be reviewed in full text form by the two reviewers independently. Conflicts will be resolved by consensus of all reviewers through a meeting. Data will be extracted by two independent reviewers using a developed data extraction tool. The results will be reported in extraction tables and diagrams with a narrative summary.

**Ethics and dissemination** This scoping review is part of the multiphase study which obtained ethical clearance from College of Medicine Research Ethics Committee in Malawi and Human Research Ethics Committee of the University of Witwatersrand, South Africa. The scoping review will be published in a peer reviewed journal. The findings will also be presented at national and international conferences.

**Trial registration number** https://doi.org/10.17605/OSF.IO/X3Q4H

## STRENGTHS AND LIMITATIONS OF THIS STUDY

⇒ The proposed scoping review will be conducted in accordance with Joanna Briggs Institute methodology for scoping reviews.
⇒ A comprehensive systematic search strategy will be conducted. Articles to be included will be those published in English within a 10-year period, that describe a paediatric oncology nursing education and training programme, from high/low/middle-income countries. This review will not report on the quality of included studies, but rather identify existing literature for paediatric oncology nursing education and training programme.
⇒ The review will map content, cultural/contextual issues, delivery methods, duration and mode of assessment for the paediatric oncology nursing education and training programmes.
⇒ Patients and the public were not involved in the design of this scoping review protocol.
⇒ The inclusion of studies published in English language only and systematic search of a few selected databases will limit the findings, thereby providing incomplete mapping of evidence.

## INTRODUCTION

Childhood cancer, also known as paediatric oncology, is a leading cause of death among children with non-communicable diseases worldwide.[1 2] Each year, more than 400 000 children are diagnosed with cancer and 80% of these are from low/middle-income countries (LMICs).[1 3] In high-income countries (HICs), the survival rate of children with all types of cancer is more than 80% while in LMICs, the survival rate ranges from 5% to 60%.[4 5] The few children that survive in LMICs live with a disability as a result of suboptimal management of the cancer.[6] Late presentation with advanced cancer disease, coupled with other factors such as abandonment of treatment, limited resources for specific cancer treatment, limited neurosurgical options, lack of radiotherapy and intense supportive care have been reported to contribute to poor survival rate of children with cancer in LMICs.[7–13]

The ability to respond to the cancer burden in LMICs has also been affected by the limited number of specialised oncology healthcare professionals, such as nurses.[14 15] Nurses form the largest group of healthcare professionals

globally and are considered the back bone of the health-care system in the management of children with cancer.[2] They provide physical, psychosocial, pharmacological and other supportive care aimed at promoting quality of life of children with cancer and their survival.[16] As such, nurses working in paediatric oncology settings need to be well prepared to perform their roles with confidence for successful team coordination and positive patient outcomes. However, the lack of nurses with paediatric oncology education and skills has been identified as a major challenge to the implementation of evidence-informed oncology nursing care.[17]

Care of children with cancer is highly specialised and requires well-educated, trained and dedicated nurses to provide high-quality care.[18] In most LMICs the under-graduate curriculum for nurses has limited content for oncology nursing, and as such nurses receive little or no paediatric oncology-specific education during pre-service education.[19] Unlike HICs and some LMICs, where nurses are given an orientation education programme when newly hired or allocated to paediatric oncology settings, many LMICs, do not have such education and training opportunities.[15 19] Nurses learn to deal with complexities of cancer management on the job through experience, intuition and doctors' instructions.[20 21] With lack of specialty training, nurses may not be familiar with aspects of care specific to patients with cancer and this may negatively affect the outcomes of care.[22]

While acknowledging that specialty education and training is the most ideal way to improve outcomes of children with cancer in LMICs, this may not be realised soon in many of these settings because the processes of developing and implementing a curriculum take a long time.[23 24] Consequently, health outcomes of children with cancer will continue to be poor as nurses managing them are not able to fully meet the specific and complex demands of paediatric cancer care due to lack of extensive knowledge and skills such as psychosocial support and safe administration of chemotherapy.[25] To address this challenge, nurses working in oncology settings in some LMICs are offered orientation education and training through a twinning programme.[26] A twinning programme is an approach where an HIC facility partners with an LMIC facility.[6 11 26] A collaborating facility from an HIC provides oncology orientation training to a partner institution in LMIC for a specific period.[11 26] However, few institutions in LMICs benefit from twinning programmes and there is no sustainability when the nurse educator returns to their home country.[27] Other factors that affect effectiveness of the twinning programmes include lack of consideration of contextual issues during the training, lack of commitment from the LMIC partner institution and the high cost associated with the twinning programme.[14] To address these challenges, Sullivan *et al*[28] recommend that hospitals from LMIC's should invest in capacity building of nurses through low-cost education programmes relevant to their context to improve patient outcomes.

Several paediatric oncology education programmes have been developed both in HICs and LMICs. These programmes have helped to prepare nurses with appropriate knowledge, skills and attitudes to become competent clinical nurses in paediatric oncology settings, thereby filling the gap of specialty education and training. The published educational programmes have used different methods of teaching such as physical class-room teaching, online learning, mentorship approach and clinical teaching.[17 19] However, no scoping review on paediatric oncology nursing education and training has been conducted. The reviews currently available focus on education offered to multidisciplinary teams,[29 30] the use of simulation-based educational strategies to educate nurses and physicians about cancer care,[31] and access to and experience of education for children and adolescents with cancer.[32] In addition, a scoping review of published training and education initiatives for health professionals in LMICs by Karim *et al*[29] was performed to understand the strategies used to train the global oncology work-force. In their review, they found that most educational initiatives targeted physicians and focused on continuing medical education. Almost all the initiatives were done under twinning programmes. From this review, one article described the impact of the initiative on patient outcomes and less than half of the initiatives involved e-learning. As such, authors recommended oncology training and education initiatives for non-physicians in LMICs.

A preliminary search for existing reviews on this topic was conducted in PubMed and Scopus databases. No protocols for a similar review were found. The current scoping review aims at describing the current literature focusing on paediatric oncology nursing education and training programmes; to map the content, delivery methods, duration and mode of assessment.

## Scoping review questions

The current scoping review aims at answering the following research questions:

1. What is the content of the paediatric oncology nursing education and training programmes?
2. What delivery methods did the paediatric oncology nursing education and training programmes use?
3. What was the duration of the paediatric oncology nursing education and training programmes?
4. How were the paediatric oncology nursing education and training programmes assessed?

## Inclusion criteria

This review will use the following inclusion criteria using the Participant, Context and Concept mnemonic.

## Participant

This review will focus on articles about paediatric oncology nursing education and training programmes. Information on paediatric oncology nursing only will be extracted from literature that combine education and training programmes for adult and paediatric oncology

nursing. For studies that focused on multidisciplinary teams, we will extract information for nurses only.

## Concept

The review will include articles that describe a paediatric oncology nursing education and training programme. The programme can be continued education, orientation training or short learning programme. Short learning programme in this review is defined as any education and training programme offered for a period of 2 hours to 6 months. This definition has been developed by the reviewers based on preliminary search and own experience with short learning programmes. Information to be extracted will include content, delivery methods, duration and mode of assessment.

## Context

The review will include articles published in peer-reviewed scientific journals and grey literature from the ProQuest Dissertation and Theses Global, paediatric oncology reports and guidelines. Articles written in English language from 2012 to 2022 will be included in the review. A 10-year period is broad enough to provide recent evidence in paediatric oncology nursing. Articles from both LMICs and HICs will be included.

## Exclusion criteria

The current review will exclude articles that report on education and training programmes for multidisciplinary teams but do not separate results for nurses and other cadres of healthcare professionals as it will be difficult to isolate data for nurses only. Articles that report on paediatric oncology nursing education and training programmes as part of a nursing curriculum for pre-service training at an academic institution such as a university will also be excluded because the focus for this scoping review is on programmes for qualified nurses.

## METHODOLOGY

The proposed scoping review will be conducted from February to September 2023 and in accordance with the Joanna Briggs Institute (JBI) methodology for scoping reviews.[33] The following steps will be followed:
1. Searching strategy.
2. Study selection.
3. Data extraction.
4. Data analysis and presentation.

### Step 1: searching strategy

A systematic search of literature using Medical Subject Headings, index terms, search terms or phrases, synonyms and alternative terms will be performed. The following search terms/phrases and their synonyms will be used: paediatric oncology nursing, childhood cancer nursing, continuing education, orientation training, educational programmes, in-service training. Search strings will be created by combining the Boolean operators OR and AND (see online supplemental appendix 1 preliminary

search strategy). The search will be conducted in the following databases: CINAHL, Dimensions, Embase, PubMed, Scopus and Education Resources Information Center. The search will aim to identify published articles from peer-reviewed journals. The search will include grey literature from the ProQuest Dissertation and Theses Global, paediatric oncology reports and guidelines as well as paediatric oncology national and international society websites. Reference lists of relevant articles will be screened to identify additional articles. An experienced librarian in systematic search of literature will adapt the attached structured search strategy to all databases depending on their design (online supplemental appendix 1).

### Step 2: study selection

Bibliographic results of the searches from all the databases including reference lists will be downloaded and uploaded into Zotero Reference Management Software for deduplication, then uploaded into Joanna Briggs Institute System for the Unified Management, Assessment and Review of Information (JBI SUMARI) for title and abstract screening. JBI SUMARI is a web-based review application that supports the steps involved in the review process from drafting a protocol, selection of studies, critical appraisal, data extraction and synthesis. It also allows one to manage review teams and contributors to the review.[34]

A two-stage standardised screening process will be employed to evaluate eligibility of the articles. Two independent reviewers (MDM, FC) will first screen the title and abstracts of the captured articles. Full text articles of the included studies will then be downloaded. Disagreements between the two reviewers will be discussed with the other three reviewers (AS, JM, IK) and final decision will be made. The results of the search will be reported in full in the final scoping review.

### Step 3: data extraction

Two independent reviewers will extract data from included articles. This will be performed using a data extraction tool developed by MDM. The data extraction tool will be tested on a sample of papers to determine its practicality prior to the data extraction process. The data to be extracted will include: author, year, journal, country, host site/institution, collaborations, location of training, project title, theory content, practical areas, cultural/contextual issues, programme delivery methods, programme duration, mode of assessment and theoretical framework underpinning the training programme.

A draft data extraction tool is provided (see online supplemental appendix 2). The draft data extraction tool will be modified and revised as necessary during the data extraction process. Modifications will be detailed in the full scoping review. Any disagreements that arise between the reviewers will be resolved through discussion or with a third reviewer.



Assessment of quality of included articles will not be performed as it is not within the scope of this review.

## Step 4: data analysis and presentation

Content analysis using basic coding of data will be performed. The results will be described narratively, and presented using summary of findings table and flow diagrams. The evidence will be summarised in relation to the purpose of this review and programmes that are relevant in LMICs will be identified.

## Ethics and dissemination

This scoping review is part of the multiphase study. The study obtained ethical clearance from Malawi, the study site (Approval number P.12/21/3537) and Human Research Ethics Committee of the University of the Witwatersrand (M220429). Findings from the scoping review will guide the third phase of the study and will contribute towards a Doctor of Philosophy in Nursing for MDM. The scoping review will be published in a peer-reviewed journal. The findings will also be presented at national and international conferences.

## Patient and public involvement

None.

## Discussion

To the best of our knowledge, this will be the first scoping review that will map available evidence on paediatric oncology nursing education and training programmes. We expect that the results of this review will be of interest to nurses practicing in different paediatric oncology settings, nursing education institutions and nursing regulatory bodies in LMICs as well as HICs. The findings will also inform the development of a paediatric oncology short learning programme for nurses in Malawi and other similar contexts.

**Acknowledgements** All authors for their contributions.

**Contributors** The review was drafted and designed by MDM. The revision was performed by FC, AS, IK, JM. All authors approved the final scoping review protocol and agreed on the submission for publication in a peer-reviewed scientific journal.

**Funding** This work is supported by the Consortium for Advanced Research Training in Africa (CARTA). CARTA is jointly led by the African Population and Health Research Center and the University of the Witwatersrand and funded by the Carnegie Corporation of New York (Grant No. G-19-57145), Sida (Grant No:54100113), Uppsala Monitoring Center, Norwegian Agency for Development Cooperation (Norad) and by the Wellcome Trust (reference no. 107768/Z/15/Z) and the UK Foreign, Commonwealth and Development Office, with support from the Developing Excellence in Leadership, Training and Science in Africa (DELTAS Africa) programme. The statements made and views expressed are solely the responsibility of the Fellow. For the purpose of open access, the author has applied a CC BY public copyright license to any Author Accepted Manuscript version arising from this submission.

**Competing interests** None declared.

**Patient and public involvement** Patients and/or the public were not involved in the design, or conduct, or reporting, or dissemination plans of this research.

**Patient consent for publication** Not applicable.

**Provenance and peer review** Not commissioned; externally peer reviewed.

peer-reviewed. Any opinions or recommendations discussed are solely those of the author(s) and are not endorsed by BMJ. BMJ disclaims all liability and responsibility arising from any reliance placed on the content. Where the content includes any translated material, BMJ does not warrant the accuracy and reliability of the translations (including but not limited to local regulations, clinical guidelines, terminology, drug names and drug dosages), and is not responsible for any error and/or omissions arising from translation and adaptation or otherwise.

**ORCID iDs**
Maureen Daisy Majamanda http://orcid.org/0000-0001-8886-3158
Felix Chisoni http://orcid.org/0000-003-1798-6446
Apatsa Selemani http://orcid.org/0000-0003-1216-6158
Irene Kearns http://orcid.org/0000-0003-2667-3353
Johanna Maree http://orcid.org/0000-0002-3153-3040

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
