## [Reviewer comments · BMJ Open]

ARTICLE DETAILS

TITLE (PROVISIONAL)	Paediatric oncology nursing education and training programmes: A scoping review protocol
AUTHORS	Majamanda, Maureen; Chisoni, Felix; Selemani, Apatosa; Kearns, Irene; Maree, Johanna

VERSION 1 – REVIEW

REVIEWER	Balneaves, Lynda University of Manitoba
REVIEW RETURNED	13-Feb-2023

GENERAL COMMENTS	This is a straight-forward, well written scoping review that focusses on identifying and summarizing paediatric oncology nursing training programs with the intent of informing future programming for LMICs. There are minor suggestions to strengthen the overall manuscript: - Page 4, line 21-22 - I'm sure this was not the intent, but the wording of this sentence suggests that without paediatric oncology training, nursing care is substandard - it may not be tailored to the care of paediatric patients, but that does not mean that it is incompetent care. Also, could you please clarify if it is the case in LMICs that without pediatric trained nurses, patients would not receive anti-cancer treatment? I believe it would instead mean they would not receive appropriate psychosocial nursing care tailored to the needs of younger patients.- Page 5, line 1-6 - I am struggling with how nursing is framed in this paragraph. It is worded in such a way that hints at a lack of professionalism within the nursing profession, learning from experience, intuition, and from other health care professions. What of nurses learning from other nurses, or building on their own competencies?- Page 6, lines 1-12 - Suggest framing the research questions in the context of nursing- Page 6, lines 40 - Framing short learning programs as ranging from 30 minutes to 6 months seems very broad. Is there a reference for this?- Page 7, lines 11-12 - A bit more explanation is needed regarding why paediatric oncology programmes embedded in curriculum in a nursing training institute would be excluded from the review. Why would they be seen as problematic?
--

	- Page 7, lines 37-38 - Recommend including subject headings along with MeSH terms given the focus on nursing and the fact that they are used predominantly in CINAHL. - Page 7, Search Strategy - Despite being mentioned later in the manuscript, grey literature is not included in the Search Strategy. It is very possible that nursing curricula focused on paediatric oncology may not be formally published in peer-reviewed literature. Suggest that grey literature is considered as part of the scoping review? - Page 8, lines 40-41 - Is there any intention to formally pilot and/or test the data extraction tool which is investigator developed? In addition, it would be worth considering adding in how culture is addressed in programmes, given the discussion in the introduction? In addition, it may also be worthwhile to include the theoretical framework that guides a curriculum as it may influence it's content and delivery style. - Page 9 - Data analysis and presentation - Will there be any attempt to synthesise the findings of the scoping review and to identify programmes that will best fit LMICs context? - Page 9, lines 21-22 - It would be useful to include a bit more details about the larger study and how the scoping review will inform it. - Appendix 1 - Will need to improve the resolution of this image, which is quite blurry. Also, as a note, the search strategy will need some work as the first search listed will be too broad with just "neoplasm" included as a search term. Minor grammatical issues:  - throughout the manuscript, ensure program/programme is used consistently - Page 4, lines 16-17 - add "way" following "ideal" - Page 4, line 37 - spelling of "twining" - Page 4, lines 59-60 - remove comma after LMICs - Page 5, lines 45-46 - capitalise Google Scholar - Page 7, line 18 - please provide a reference for JBI methodology
--	---

REVIEWER	Colomer-Lahiguera , Sara Lausanne University Hospital
REVIEW RETURNED	20-Mar-2023

GENERAL COMMENTS	Paediatric oncology nursing education and training programs: A scoping review protocol Reviewer: minor revisions Comment to the authors: The scoping review protocol addresses an important issue in paediatric oncology as is the formation of paediatric oncology nurses. Mapping the different training programmes that exists will be a valuable resource. There is a focus throughout the text on LMIC. Address this in the discussion (see comment below) Introduction: Maybe paragraph 4 ("Care of children with cancer is highly specialised..." could be after paragraph 2 as it emphasizes the
---

	same idea on the need for special training. And paragraph 3 introduces the training programmes that continue in paragraph 5 “Several paediatric oncology education programmes...” ? Inclusion criteria >> Context: “Articles could also be from any setting or geographical location”. Please specify what you understand by “any setting” I would consider as well International and National Societies, as they sometimes provide also training. Will the authors focus only on training programmes in English? Or are they considering other languages? Please specify. Exclusion criteria: “The current review will exclude articles that report on education and training programmes for multidisciplinary teams but do not separate results for nurses and other cadres of health care professionals.” Why not to consider these programmes as a separate bucket of information? Authors will have identified these programmes, so instead of discarding them I would suggest to list them, without extracting data, but might be important for other researchers or groups in the future. “Articles that report on paediatric oncology nursing education and training programmes as part of a curriculum for a nursing training institution will also be excluded.” Can the authors provide examples of what they consider “programmes as part of a curriculum for a nursing training institution”. Is “institution” for instance Universities, Hospitals, National Societies...? Step 1: Searching strategy “The search will aim to identify published articles from peer reviewed journals.” Was any initial limited search conducted previously? How feasible is to identify such training programmes via peer-reviewed journals? Did the authors also consider Professional journals? And databases such ERIC (ERIC - Education Resources Information Center) https://eric.ed.gov/ Discussion: The authors could add a sentence on practical implications (will this serve to develop a specific programme in the future?) Limitations Include some sentence about potential limitations (language, databases, difficulty to have a complete map....)
--	--

VERSION 1 – AUTHOR RESPONSE

Reviewer 1: Dr. Lynda Balneaves

Comment 1

- Page 4, line 21-22 - I'm sure this was not the intent, but the wording of this sentence suggests that without paediatric oncology training, nursing care is substandard - it may not be tailored to the care of paediatric patients, but that does not mean that it is incompetent care. Also, could you please clarify if it is the case in LMICs that without pediatric trained nurses, patients would not receive anti-cancer treatment? I believe it would instead mean they would not receive appropriate psychosocial nursing care tailored to the needs of younger patients.

Response 1

Page 4

This has been addressed and the sentence reads as follows:

Consequently, children are at risk of dying of cancer as they continue to be managed by untrained nurses who are not able to fully meet the demands of pediatric cancer care due to lack of extensive knowledge and skills such as psychosocial support²⁴.

Comment 2

- Page 5, line 1-6 - I am struggling with how nursing is framed in this paragraph. It is worded in such a way that hints at a lack of professionalism within the nursing profession, learning from experience, intuition, and from other health care professions. What of nurses learning from other nurses, or building on their own competencies?

Response 2

The paragraph has been addressed as follows:

Nurses learn to deal with complexities of cancer management on the job through experience, intuition and doctors' instructions ^{20,21}. With lack of speciality training, nurses are not able to effectively manage children with cancer and this negatively affects the outcomes of care.

The statement has been moved to page 4 paragraph 2 as suggested by the second reviewer.

Comment 3

- Page 6, lines 1-12 - Suggest framing the research questions in the context of nursing

Response 3

Page 6

The questions have been framed in the context of nursing as follows:

1. What is the content of the paediatric oncology nursing education and training programmes?
2. What delivery methods did the paediatric oncology nursing education and training programmes use?
3. What was the duration of the paediatric oncology nursing education and training programmes?
4. How was the paediatric oncology nursing education and training programmes assessed?

Comment 4

- Page 6, lines 40 - Framing short learning programs as ranging from 30 minutes to 6 months seems very broad. Is there a reference for this?

Response 4

With the preliminary search, it was found that some short learning programmes focused on one short topic which was given for two hours only, other programmes used the high dose low intensity approach (this is an intense programme for a short period of time) and these took a duration of one day to four weeks while others used a low dose high intensity approach where a few hours per week were allocated for the training programme but spread over a five to six months' period.

The sentence has been refined and reads as follows:

Short learning program in this review is defined as any education and training program offered for a period of two hours to six months.

This definition has been developed by the reviewers based on preliminary search and own experience with short learning programmes and no reference has been accessed on this definition.

Comment 5

- Page 7, lines 11-12 - A bit more explanation is needed regarding why paediatric oncology programmes embedded in curriculum in a nursing training institute would be excluded from the review. Why would they be seen as problematic?

Response 5

Page 7

This section has been edited as follows:

Articles that report on paediatric oncology nursing education and training programmes as part of a nursing curriculum for pre service training at an academic institution such as a university will also be

excluded because the focus for this scoping review is on programmes for qualified nurses.

Comment 6

- Page 7, lines 37-38 - Recommend including subject headings along with MeSH terms given the focus on nursing and the fact that they are used predominantly in CINAHL.

Response 6

Page 7

Instead of Subject headings we have used its synonym in the search strategy and the sentence reads as follows:

A systematic search of literature using Medical Subject Headings (MeSH), index terms, search terms or phrases, synonyms and alternative terms will be performed.

Comment 7

- Page 7, Search Strategy - Despite being mentioned later in the manuscript, grey literature is not included in the Search Strategy. It is very possible that nursing curricula focused on paediatric oncology may not be formally published in peer-reviewed literature. Suggest that grey literature is considered as part of the scoping review?

Response 7

Page 7

The search strategy has been revised and now includes grey literature. The sentence is now reading as follows:

The search will include grey literature from the ProQuest Dissertation and Theses Global, paediatric oncology reports and guidelines as well as paediatric oncology national and international society websites.

Comment 8

- Page 8, lines 40-41 - Is there any intention to formally pilot and/or test the data extraction tool which is investigator developed? In addition, it would be worth considering adding in how culture is addressed in programmes, given the discussion in the introduction? In addition, it may also be worthwhile to include the theoretical framework that guides a curriculum as it may influence its content and delivery style.

Response 8

There is a plan to test the pre developed data extraction tool. We have included this statement on page 8 and reads as follows:

The data extraction tool will be tested on a sample of papers to determine its practicality prior to the data extraction process.

On page 8, we have added cultural/ contextual issues and theoretical framework underpinning the training programme as part of data to be extracted.

Comment 9

- Page 9 - Data analysis and presentation - Will there be any attempt to synthesise the findings of the scoping review and to identify programmes that will best fit LMICs context?

Response 9

We will synthesise the findings of the scoping review. A sentence on this has been included on step 4 page 9 as follows:

The evidence will be summarised in relation to the purpose of this review and programmes that are relevant in LMICs will be identified.

Comment 10

- Appendix 1 - Will need to improve the resolution of this image, which is quite blurry. Also, as a note, the search strategy will need some work as the first search listed will be too broad with just "neoplasm" included as a search term.

Response 10

Search terms have been inserted into a Word document (Appendix 1) as it remained blurry on the screenshot.

Yes, the term “neoplasm” is too broad but the results will be narrowed down due to a combination with the other search concepts in Search #4. This term has also been used in a similar context elsewhere, see <https://journals.plos.org/globalpublichealth/article?id=10.1371/journal.pgph.0000098>

Comment 11

Minor grammatical issues:

- throughout the manuscript, ensure program/programme is used consistently
- Page 4, lines 16-17 - add "way" following "ideal"
- Page 4, line 37 - spelling of "twining"
- Page 4, lines 59-60 - remove comma after LMICs
- Page 5, lines 45-46 - capitalise Google Scholar
- Page 7, line 18 - please provide a reference for JBI methodology

Response 11

-We have replaced program with programme in the document.

-Page 4 paragraph 3 the word way has been added after ideal and the sentence reads as follows:

While acknowledging that specialty education and training is the most ideal way to improve outcomes of children with cancer in LMICs.

-Page 5 the spelling of twining has been corrected to twinning.

-The comma after LMICs has been removed.

-Page 5 Google Scholar has been capitalised.

-Page 7 Reference for JBI methodology has been provided.

Reviewer 2: Dr. Sara Colomer-Lahiguera

Comment 1

There is a focus throughout the text on LMIC. Address this in the discussion (see comment below)

Introduction:

Maybe paragraph 4 (“Care of children with cancer is highly specialised...” could be after paragraph 2 as it emphasizes the same idea on the need for special training. And paragraph 3 introduces the training programmes that continue in paragraph 5 “Several paediatric oncology education programmes...” ?

Response 1

Paragraph 4 has been moved up and is now paragraph 3 on page 4.

Comment 2

Inclusion criteria >> Context: “Articles could also be from any setting or geographical location”. Please specify what you understand by “any setting”

Response 2

Setting in this case means articles could also be from LMICs and HICs. This has been addressed on page 7 under context as follows: Articles could also be from LMICs and HICs.

Comment 3

I would consider as well International and National Societies, as they sometimes provide also training.

Response 3

Page 7

The search strategy has been revised and now includes paediatric oncology national and international society websites. The sentence is now reading as follows:

The search will include grey literature from the ProQuest Dissertation and Theses Global, paediatric oncology reports and guidelines as well as paediatric oncology national and international society

websites.

Comment 4

Will the authors focus only on training programmes in English? Or are they considering other languages? Please specify.

Response 4

The authors will consider articles published in English only because we do not have the capacity for interpretation of other languages.

This has been specified on page 6 lines 54-55 as follows:

Articles written in English language from 2012 to 2022 will be included in the review.

Comment 5

Exclusion criteria:

“The current review will exclude articles that report on education and training programmes for multidisciplinary teams but do not separate results for nurses and other cadres of health care professionals.”

Why not to consider these programmes as a separate bucket of information? Authors will have identified these programmes, so instead of discarding them I would suggest to list them, without extracting data, but might be important for other researchers or groups in the future.

Response 5

The programmes for multidisciplinary teams will be used as evidence for the findings from the included articles in the discussion section and will be referenced accordingly. We find that listing them when we will not extract data may be confusing to the readers.

Comment 6

“Articles that report on paediatric oncology nursing education and training programmes as part of a curriculum for a nursing training institution will also be excluded.”

Can the authors provide examples of what they consider “programmes as part of a curriculum for a nursing training institution”. Is “institution” for instance Universities, Hospitals, National Societies...?

Response 6

Page 7

This section has been edited as follows:

Articles that report on paediatric oncology nursing education and training programmes as part of a nursing curriculum for pre service training at an academic institution such as a University will also be excluded because the focus for this scoping review is on programmes for qualified nurses.

Comment 7

Step 1: Searching strategy

“The search will aim to identify published articles from peer reviewed journals.”

Was any initial limited search conducted previously? How feasible is to identify such training programmes via peer-reviewed journals?

Response 7

Yes, a limited search was conducted across major databases such as PubMed, CINAHL and Scopus and grey literature sources but also reference searching of the most relevant papers that were initially sourced. It was therefore found feasible that it would be possible to find relevant studies for this review through peer-reviewed journals.

Comment 8

Did the authors also considered Professional journals? And databases such ERIC (ERIC - Education Resources Information Center) <https://eric.ed.gov/>

Response 8

Page 7

This has been considered under step 1 search strategy as follows:

The search will be conducted in the following databases: CINAHL, Dimensions, Embase, PubMed, Scopus and Education Resources Information Center (ERIC). The search will aim to identify published articles from peer reviewed journals. The search will include grey literature from the ProQuest Dissertation and Theses Global, paediatric oncology reports and guidelines as well as paediatric oncology national and international society websites.

Comment 9

Discussion:

The authors could add a sentence on practical implications (will this serve to develop a specific programme in the future?)

Response 9

Page 10

The following sentence has been added under discussion:

The findings will also inform the development of a paediatric oncology short learning programme for nurses in Malawi and other similar context.

Comment 10

Limitations

Include some sentence about potential limitations (language, databases, difficulty to have a complete map....)

Response 10

Limitations section has been upgraded as follows: Inclusion of studies published in English language only and searching for articles in few databases will limit the findings thereby providing incomplete mapping of the evidence.

VERSION 2 – REVIEW

REVIEWER	Colomer-Lahiguera , Sara Lausanne University Hospital
REVIEW RETURNED	05-May-2023

GENERAL COMMENTS	There are strong statements in paragraphs 3 and 4 of the introduction that lack evidence. I think the add-ons do not respond to what was pointed by the reviewer and rather have the opposite effect, questioning basic nursing competencies. Furthermore, I don't think citation 24 is used correctly in this sentence. The paper measures pediatric oncology nursing education outcomes, but no survival. Therefore, authors should avoid statements such as "Consequently, children are at risk of dying of cancer as they continue to be managed by untrained nurses". Please revise the introduction to avoid such statements. Corrections were highlighted in red, however deleted sentences should also appear as strikethrough text for change tracking. Answer 2 for Reviewer 2: Writing "Articles could also be from LMICs and HICs" instead of "Articles could also be from any setting or geographical location". The way it has been corrected is redundant with the last sentence of "Participant" in the Context section.
---

VERSION 2 – AUTHOR RESPONSE

Reviewer 1 queried "Also, could you please clarify if it is the case in LMICs that without pediatric trained nurses, patients would not receive anti-cancer treatment? I believe it would instead mean they would not receive appropriate psychosocial nursing care tailored to the needs of younger patients." Please ensure this is addressed in the manuscript and response letter.	This has been addressed as follows: Under introduction, paragraph number four from line 125 Consequently, health outcomes of children with cancer will continue to be poor as nurses managing them are not able to fully meet the specific demands of pediatric cancer care due to lack of extensive knowledge and skills such as psychosocial support and safe administration of chemotherapy²⁴	
6	In response to Reviewer 1, you added "With lack of speciality training, nurses are not able to effectively manage children with cancer and this negatively affects the outcomes of care." to the manuscript, please include a reference for this statement.	Reference for this statement has been added. It is citation number 22 on page 4.
7	In response to Reviewer 1, you clarified that "This definition has been developed by the reviewers based on preliminary search and own experience with short learning programmes and no reference has been accessed on this definition." Please include this in the manuscript.	This statement has been added in the manuscript and reads as follows under context from line 202: Short learning programme in this review is defined as any education and training programme offered for a period of two hours to six months. This definition has been developed by the reviewers based on preliminary search and own

		experience with short learning programmes.
8	Please work to improve the quality of the writing throughout your manuscript. We recommend asking a colleague who is proficient in written English to assist you; alternatively, you could enlist the help of a professional copyediting service.	After addressing all comments from reviewers, we sent the document to our colleague who is proficient in written English for proof reading and has he has worked on the whole document.
	Reviewer 2	
1	There are strong statements in paragraphs 3 and 4 of the introduction that lack evidence. I think the add-ons do not respond to what was pointed by the reviewer and rather have the opposite effect, questioning basic nursing competencies.	We have revised the statement that was questioning basic nursing competencies to focus on lack of specific skills in paediatric oncology from line number 125. The new sentence reads as follows and has a reference: Consequently, health outcomes of children with cancer will continue to be poor as nurses managing them are not able to fully meet the specific and complex demands of pediatric cancer care due to lack of extensive knowledge and skills such as psychosocial support and safe administration of chemotherapy.
2	Furthermore, I don't think citation 24 is used correctly in this sentence. The paper measures pediatric oncology nursing education outcomes, but no survival. Therefore, authors should	Previous citation 24 was Day et al 2012.

	avoid statements such as “Consequently, children are at risk of dying of cancer as they continue to be managed by untrained nurses”. Please revise the introduction to avoid such statements.	And the citation has been replaced with Uwayezu et al 2022. And the introduction has been revised. The statement mentioned in the comments has been revised as follows: Consequently, health outcomes of children with cancer will continue to be poor as nurses managing them are not able to fully meet the specific and complex demands of pediatric cancer care due to lack of extensive knowledge and skills such as psychosocial support and safe administration of chemotherapy
3	Corrections were highlighted in red, however deleted sentences should also appear as strikethrough text for change tracking.	All changes and new additions are in track changes now.
4	Answer 2 for Reviewer 2: Writing “Articles could also be from LMICs and HICs” instead of “Articles could also be from any setting or geographical location”. The way it has been corrected is redundant with the last sentence of “Participant” in the Context section.	Thank you so much for this observation. There was redundancy indeed. The information in the last sentence under participant has been removed.